# Understanding Cerebellar Input Stage through Computational and Plasticity Rules

**DOI:** 10.3390/biology13060403

**Published:** 2024-06-01

**Authors:** Eleonora Pali, Egidio D’Angelo, Francesca Prestori

**Affiliations:** 1Department of Brain and Behavioral Sciences, University of Pavia, 27100 Pavia, Italy; eleonora.pali01@universitadipavia.it (E.P.);; 2Digital Neuroscience Center, IRCCS Mondino Foundation, 27100 Pavia, Italy

**Keywords:** cerebellum, plasticity, granule cells, Golgi cells, unipolar brush cells

## Abstract

**Simple Summary:**

One of the main theories regarding brain functioning is that plasticity modifies the effectiveness of synaptic transmission to control the signal transfer function. It has been demonstrated that the granular layer of the cerebellum regulates the gain of signals that travel via the mossy fiber channel. Until now, the influence of plasticity on incoming activity patterns has been examined through a combination of computational modeling and electrophysiological recordings in cerebellar slices. This approach revealed a wide range of distinct types of synaptic plasticity in the granular layer, frequently in conjunction with changes in the intrinsic excitability. Here, we provide a quick summary of the most common types of plasticity observed at the excitatory synapses formed by mossy fibers onto principal neurons in the granular layer. Next, we emphasize our current understanding of the mechanisms behind synaptic and intrinsic plasticity and how they affect function, offering important new perspectives on how information is interpreted and rearranged at the cerebellar input stage.

**Abstract:**

A central hypothesis concerning brain functioning is that plasticity regulates the signal transfer function by modifying the efficacy of synaptic transmission. In the cerebellum, the granular layer has been shown to control the gain of signals transmitted through the mossy fiber pathway. Until now, the impact of plasticity on incoming activity patterns has been analyzed by combining electrophysiological recordings in acute cerebellar slices and computational modeling, unraveling a broad spectrum of different forms of synaptic plasticity in the granular layer, often accompanied by forms of intrinsic excitability changes. Here, we attempt to provide a brief overview of the most prominent forms of plasticity at the excitatory synapses formed by mossy fibers onto primary neuronal components (granule cells, Golgi cells and unipolar brush cells) in the granular layer. Specifically, we highlight the current understanding of the mechanisms and their functional implications for synaptic and intrinsic plasticity, providing valuable insights into how inputs are processed and reconfigured at the cerebellar input stage.

## 1. Introduction

Long-lasting changes in synaptic connectivity within neural networks are key to shaping the activity of neuronal populations [1,2]. Despite being overlooked in classical cerebellar theories [3,4], empirical studies and computational modeling have unraveled a broad spectrum of synaptic and intrinsic plasticity mechanisms in the granular layer, providing valuable insights into how inputs are processed and reconfigured at the cerebellar input stage.

## 2. Plasticity in Granule Cells

A multitude of inputs from various brain regions are conveyed via mossy fibers into the granular layer [5,6,7]. Within this layer, the mossy fibers branch out across different folia, creating multiple ramifications. Each branch then gives rise to numerous rosettes, which are central components of the cerebellar glomeruli [6,8]. These rosettes consist of presynaptic elements characterized by multilobed grooves, where dendrites from tens of granule cells establish contact. The cerebellar glomerulus is completed by the axons of Golgi cells and their basal dendrites, which receive input from mossy fibers and ascending axons of granule cells [9]. Over the past few decades, studies have revealed that prolonged high-frequency discharges of mossy fibers, extending beyond a specific duration, induce long-term potentiation (LTP) at the mossy fiber–granule cell synapse [10,11]. The induction of mossy fiber–granule cell LTP depends on intricate changes in the intracellular calcium concentrations, orchestrated by a complex interplay of key factors. First, NMDA receptors (NMDARs) serve as the primary conduits for Ca^2+^ influx, together with metabotropic glutamate receptors 1 (mGluR1s), which enhance this process via the inositol trisphosphate (IP_3_) intracellular pathway [12,13]. Secondly, activation of voltage-dependent calcium channels (VDCCs) prompts membrane depolarization, and the generation of repetitive spike discharges, significantly contributing to the LTP induction [10]. Thirdly, intracellular Ca^2+^ signal modulation through Ca^2+^-induced Ca^2+^ release (CICR) contributes to the long-term plasticity by amplifying and prolonging the calcium signals [14,15,16]. At different synapses, LTP controls numerous functional aspects of the synapse, including neurotransmitter release, spillover, and postsynaptic receptor gating and expression [17,18,19]. Based on patch-clamp recordings and mathematical modeling, the mossy fiber–granule cell LTP is characterized by an increased probability of neurotransmitter release [20,21]. However, since the discovery of LTP at the mossy fiber–granule cell synapse, our understanding of the plasticity mechanisms in cerebellar granule cells has evolved, uncovering a new layer of complexity (Figure 1).

### 2.1. LTP/LTD Balance at Mossy Fiber–Granule Cell Synapse

According to the Hebb postulate [25] and its extension in the Bienenstock–Cooper–Munro model (BCM; [26]), both LTP and LTD can characterize a specific synapse. Their induction depends on varying levels of postsynaptic [Ca^2+^]_i_ increase via the activation of distinct biochemical pathways [27,28]. Ex vivo and in vivo studies of cerebellar plasticity have elucidated a bidirectional modulation involving both LTP and LTD at the mossy fiber–granule cell relay [22,23,29]. The balance between LTP and LTD is crucial for supporting computation and learning in the granular layer, and its regulation relies on diverse mechanisms. Intracellular postsynaptic calcium levels can be regulated by burst patterns of mossy fiber discharge. The extent of the [Ca^2+^]_i_ increase correlates with the duration of the mossy fiber bursts, resembling a BCM-like relationship [26]. Short, isolated bursts with minor [Ca^2+^]_i_ fluctuations induce LTD, whilst prolonged or repeated bursts leading to a significant [Ca^2+^]_i_ increase result in LTP (Figure 1A). Bidirectional long-term synaptic plasticity can also be influenced by the frequency-coded pattern of mossy fiber stimulation, resulting in different levels of postsynaptic elevations in [Ca^2+^]_i_ (Figure 1B). Interestingly, this relationship between the plasticity and [Ca^2+^]_i_ levels mirrors that induced by high-frequency bursts of varying durations, except that, in this case, low-frequency LTD necessitates the involvement of mGluRs rather than NMDARs. This indicates that the receptor pathways activated by diverse induction patterns via the calcium concentration change the shared plastic mechanisms to promote the reconfiguration of inputs within the granular layer [22,23]. However, the balance between LTP and LTD at the mossy fiber–granule cell synapse is not only orchestrated by specific input patterns but also by molecular factors and neuromodulators. Nitric oxide (NO) is released in the granular layer upon high-frequency mossy fiber stimulation via NMDAR-dependent and NOS-dependent mechanisms. Acting as a retrograde messenger, NO favors the increase of presynaptic release probability, thus promoting LTP over LTD [30]. Furthermore, the activation of α7 nicotinic acetylcholine receptors (α7nAchRs) on both mossy fiber terminals and granule cell dendrites amplifies the influx of Ca^2+^ at postsynaptic sites. This intensified Ca^2+^ influx has the potential to shift LTD toward LTP, sustaining this plasticity for prolonged durations [31]. In addition, recent in vivo studies have proposed that 20 Hz facial stimulation induces mossy fiber–granule cell LTP via the NMDA receptor/NO signaling pathway, which is significantly enhanced by nicotine [32,33]. Specifically, blockade of NMDA receptor-dependent mossy fiber–granule cell LTP has revealed a nicotine-triggered plasticity through nAChR activation [33]. Notably, this facial stimulation-induced LTP of mossy fiber–granule cell synaptic transmission is abolished by inhibiting NOS, regardless of the absence or presence of nicotine. This suggests that NOS activation is not only required for the induction of facial stimulation-induced mossy fiber-granule cell LTP under control conditions but also for the nicotine-induced enhancement of mossy fiber–granule cell synaptic transmission LTP [32,33].

### 2.2. Spike-Timing-Dependent Plasticity at Mossy Fiber–Granule Cell Synapse

Expanding upon the BCM theory, spike-timing-dependent plasticity (STDP) represents a unique form of synaptic plasticity where the temporal sequence of the presynaptic and postsynaptic spikes dictates the postsynaptic [Ca^2+^]_i_ levels, determining whether a synapse is strengthened or weakened [34,35,36,37,38,39,40,41,42,43]. A recent finding has revealed that the interplay between the synaptic response (EPSP) and spikes induced without synaptic activation can drive STDP at the mossy fiber–granule cell synapse ([24]; Figure 1C). Additionally, the activation of GABA_A_ receptors has been shown to reverse the phase dependence of this STDP, exhibiting an anti-Hebbian effect. Although the molecular mechanisms underlying GABAergic regulation remain elusive, this phenomenon offers a potent mechanism for diversifying the array of possible STDP configurations at mossy fiber–granule cell synapses. Furthermore, mossy fiber–granule cell STDP has been found to be tuned to the theta band (6 Hz) at which Golgi cells and granule cells show oscillations [44]. Therefore, the mossy fiber–granule cell STDP may play a crucial role in linking cerebellar learning with the low-frequency oscillations observed in thalamocortical and hippocampal circuits, facilitating the coordination of learning and memory processes during different functional states, such as voluntary movement, resting attentiveness, and sleep [45,46,47,48,49].

### 2.3. Granule Cell Intrinsic Plasticity

The regulation of information processing in the brain also encompasses persistent adjustments in the ionic conductances within specialized neuronal regions, influencing the long-term propagation of neuronal information [50,51]. Following high-frequency activation of the mossy fiber–granule cell relay, LTP triggers a sustained enhancement in granule cell electroresponsiveness, intensifying their firing (Figure 1D). This enhancement involves an increase in the granule cell input resistance and a reduction in the spike threshold, possibly due to changes in the persistent sodium and potassium currents [10,20]. Moreover, similar to synaptic LTP, the intrinsic plasticity in granule cells relies on NMDAR activation, ensuring a specific rise in the intracellular calcium concentration and subsequent activation of the calcium-dependent intracellular pathways. Intrinsic plasticity can restore the granule cell excitability levels when synaptic excitation is weak [10]. Through this plastic mechanism, the low background firing rates of granule cells, resulting from tonic inhibition by Golgi cells, can thus be reinstated by precise mossy fiber activation, inducing bursting in granule cells [10,52]. Therefore, this mechanism ensures high reliability in transmitting specific mossy fiber input patterns to Purkinje cells, thereby regulating input processing and phase learning in the cerebellum. Additionally, when local groups of mossy fibers are active, the pattern of granule cell firing in a region will be determined by the strength of the mossy fiber connections, synchronized quantal release, and improved reliability of the excitatory drive, increasing the fraction of active granule cells. This may allow temporal correlations across multiple mossy fibers to be transmitted through the granular layer as intense patches of synchronized activity, while allowing rate-coded signals to be mediated by lower intensity, uncorrelated granule cell activity [53].

## 3. Plasticity in Golgi Cells

In recent decades, extensive research has shed light on the vital role played by Golgi cells in orchestrating the activity of granule cells [54,55]. Functioning as the primary source of inhibition for granule cells, Golgi cells establish two essential inhibitory circuits: feedforward and feedback inhibition. Feedforward inhibition occurs when mossy fibers activate Golgi cells in the glomerulus, leading to subsequent inhibition of granule cells via their axons. The GABAergic synapses formed by Golgi cells onto granule cells operate through two mechanisms: phasic and tonic inhibition [9]. Phasic inhibition, mediated by synaptic GABA_A_ receptors, enhances the precision of granule cell spike-timing by narrowing the time window for synaptic integration (for the time window effect, see [56]). Tonic inhibition, mediated by extrasynaptic GABA_A_ receptors, establishes the baseline level of granule cell excitability, allowing the neuron to discern significant information from background noise. Feedback inhibition occurs when Golgi cells send back inhibitory signals to the granule cells in response to an excitatory stimulus received from nearby granule cells [9]. Through these two inhibitory loops, Golgi cells regulate the balance between excitation and inhibition, with high excitatory/inhibitory (E/I) ratios promoting LTP, intermediate ratios inducing LTD, and very low ratios preventing plastic changes, thereby serving as a primary regulator of long-term synaptic plasticity within the granular layer circuit [56]. Moreover, Golgi cells, with their extensive axonal plexus, establish lateral inhibition, resulting in the center-surround (C/S) organization of granule cell activity [57]. Under prolonged high-frequency activation of the mossy fiber bundle, the granular layer network exhibits specific spatial organization, where high excitation levels in the core permit the passage of high-frequency discharges, while inhibition in the surround organization filters out low-frequency discharges. This facilitates the transmission of high-frequency bursts along channels formed by the ascending axons of granule cells heading toward the molecular layer [58]. As a result, LTP and LTD are organized in C/S structures, with more active centers favoring LTP and less active surrounds preferentially inducing LTD, thus accelerating (LTP) or delaying (LTD) the granule cell responses to mossy fiber bursts [57]. Overall, Golgi cells play a crucial role in controlling the information flow to Purkinje cells by shaping granule cell activity: feedforward inhibition introduces a time-windowing effect, lateral inhibition establishes a center-surround organization of granular cell layer responses, and local control of the granule cell excitatory–inhibitory balance determines inhibition-controlled plasticity [59]. Furthermore, feedback granule cell inhibition, combined with the broad convergence over numerous Golgi cells, forms the substrate for generating regular synchronous oscillations over large granular cell layer fields [44,45]. Although Golgi cells’ function in granule cells’ inhibition-controlled plasticity has been thoroughly studied, little is known about the synaptic and intrinsic plasticity of Golgi cells. However, recent investigations and mathematical models of cerebellar Golgi cells have unveiled critical plastic mechanisms that could serve as potent regulatory mechanisms of granular layer plasticity (Figure 2).

### 3.1. LTP/LTD Balance at the Mossy Fiber–Golgi Cell Synapse

A recent study has unveiled bidirectional plasticity at mossy fiber–Golgi cell synapses, marking the first observation of such phenomena in cerebellar Golgi cells [60]. This synaptic plasticity demonstrates a unique voltage dependence, with the direction of plasticity (LTD or LTP) determined by the membrane potential of Golgi cells during theta-burst stimulation (TBS) induction. Specifically, LTP is favored when TBS occurs at depolarized Golgi cell potentials, while LTD is favored at hyperpolarized potentials (Figure 2A). The mechanisms underlying this voltage-dependent plasticity involve the activation of distinct calcium channels: LTP requires the activation of T-type Ca^2+^ channels alongside NMDARs, whereas LTD uniquely relies on L-type Ca^2+^ channels [60]. Notably, this voltage-dependent plasticity differs from the purely NMDAR-dependent plasticity observed at neighboring mossy fiber–granule cell synapses, suggesting that mossy fiber presynaptic terminals engage diverse induction mechanisms depending on the target cell. Furthermore, considering that multiple long-term regulatory mechanisms may coexist, dictating the balance of plasticity at the mossy fiber–Golgi cell synapse, a recent realistic model of Golgi cells predicted that temporally correlated mossy fiber–parallel fiber inputs can induce spike-timing-dependent plasticity (STDP) at the mossy fiber–Golgi cell synapse when NMDAR is activated ([61]; Figure 2B). Specifically, the integration of multimodal information from parallel fibers by apical dendrites could control the coincidence of spikes with specific mossy fiber inputs arriving on basal dendrites, thereby driving the shift between LTD and LTP at the mossy fiber–Golgi cell relay. Therefore, with their electrogenic architecture, Golgi cells may act as cortical detectors, finely regulating the information flow through the granular layer under parallel fiber with high temporal precision. Considering that spike timing has been demonstrated to control the LTP/LTD balance at neighboring mossy fiber–granule cell synapses [24], the STDP phenomenon might extend to the entire granular layer, supporting the cerebellar capacity to sustain input integration and network functioning. According to a theoretical model [63], the granular layer circuit may indeed operate within a learning framework. In this scheme, incoming mossy fiber information could be initially stored in the oscillatory network of Golgi cells, subsequently driving STDP at neighboring mossy fiber–granule cell synapses, thus orchestrating spatiotemporal input processing and plasticity within the granular layer.

### 3.2. Golgi Cell Intrinsic Plasticity

Golgi cell activity can also be affected by other mechanisms that can alter inhibitory transmission and its impact on granule cell plasticity. Transient hyperpolarization of Golgi cells leads to a significant, long-term increase in their spontaneous firing rate, a phenomenon known as firing rate potentiation (FRP) (Figure 2C). The FRP in Golgi cells depends on electrical couplings that regulate the timing and extent of both spontaneous and sensory-evoked correlated activities. This regulation involves cooperative effects related to shared synaptic depolarization, spikelet transmission, and plasticity mechanisms. Furthermore, this process is mediated by calcium-calmodulin-dependent kinase II (CaMKII) and BK-type calcium-activated potassium channels [62]. Golgi cell inhibitory activity is also regulated by gap junctions [64,65,66]. Within this electrically coupled network of Golgi cells, action potential synchronization occurs in the absence of correlated input, while transient desynchronization occurs with sparse excitatory synaptic input [44,65]. The amplification of the gap junction signals, attributed to sodium currents, implies that electrical synapses are not merely passive intercellular channels but rather dynamic forms of interneuronal communication [66]. Therefore, like chemical synapses, the electrical ones can also exhibit plasticity and be modifiable [67,68]. This introduces new perspectives on the concept of gap junction plasticity, a homeostatic mechanism recently explored in various brain regions [69,70,71,72]. A computational model investigating this type of plasticity within a recurrent cortical network [72] simulated that it can regulate the balance between synchronous and asynchronous patterns of activity in the network. Strong electrical coupling between neurons tends to induce oscillatory activities characterized by synchronized bursting mediated by inhibitory neurons. These bursts trigger the depression of the gap junctions, allowing the network to transition away from the oscillatory regime and spike asynchronously. Conversely, in the asynchronous regime, neuronal firing is sparse, contributing to the potentiation of gap junctions. Consequently, the irregular regime tends to strengthen the connections between neurons via gap junctions. This suggests a direct functional role for gap junction plasticity in information transmission within neuronal assemblies [69,70,71,72]. However, despite evidence of modifiable electrical synapses and oscillations in Golgi cells, gap junction plasticity remains unexplored and warrants further investigation.

## 4. Plasticity in Unipolar Brush Cells

Unipolar brush cells (UBCs) are glutamatergic interneurons primarily located in the granular layer of the vestibulocerebellum [73,74,75]. Unlike granule cells, which typically possess 4–5 dendrites innervated by various mossy fibers, UBCs have a single short dendrite with a brush-like appearance, typically receiving inputs from a single mossy fiber terminal [73]. Additionally, they possess axonal branches constituting a noncanonical, cortex-intrinsic subset of mossy fibers that form synapses with both granule cells and other UBCs [74,76,77,78]. Consequently, while neurotransmitters generally have a short lifespan in the synaptic cleft due to their rapid diffusion, the specialized three-dimensional arrangement of mossy fiber–UBC synapses promotes a persistent presence of glutamate in the cleft [79,80]. This leads to prolonged and repetitive activation of postsynaptic receptors in UBCs. As a result, in response to mossy fiber input, UBCs display a unique dual-component response: a rapid initial AMPA receptor-mediated current, followed by a sustained tail of inward current lasting seconds (i.e., steady-state current) [78,81,82,83,84]. Phasic excitation can induce robust action potential bursts and transiently elevated firing rates, whilst persistent inward currents in UBCs facilitate periods of tonic action potential firing. Short-term plasticity mechanisms further refine and modulate both transient and prolonged synaptic currents in UBCs. Fast AMPA and kainate receptor-mediated responses exhibit the depression at short inter-stimulus intervals, characterized by a prolonged reduction in the amplitude of the fast excitatory postsynaptic currents (EPSCs) evoked by a secondary mossy fiber stimulus [79,80,84]. This short-term depression reflects the increased availability of unbound AMPA receptors over time, resulting in a smaller response to the second stimulus. Conversely, the responses of the slower steady-state current can be either facilitated or depressed, depending on the inter-stimulus interval duration. Specifically, at short intervals (50–400 ms), the initial steady-state current exhibits a positive undershoot, reversing between 400 and 600 ms, and gradually recovering to negative values after several seconds [80]. The characteristics of postsynaptic receptors, along with the limitations on glutamate diffusion due to synaptic ultrastructure and glutamate transporters, together influence the time course of the resulting steady-state current. In response to individual presynaptic inputs, this causes a prolonged sequence of action potentials in UBCs.

### The Impact of UBC on Temporal Dynamics in the Granular Layer Network

The cerebellar ability to learn complex input–output relationships across various time scales relies on establishing a diverse temporal framework [56,85,86,87,88,89]. Consequently, mossy fiber inputs undergo intricate transformations within the granular layer, giving rise to highly complex spatial-temporal activity patterns in granule cells. Central to this process are UBCs, which play essential roles in shaping the activity dynamics of the granule cell population. UBCs extend axonal branches within the granular layer, likely innervating a group of granule cells [78,79]. A single action potential in UBCs may thus initiate a cascade of EPSCs across a large ensemble of granule cells, profoundly impacting their activity [84]. Consequently, UBCs ensure the amplification of mossy fiber signals in a feedforward manner, enabling complex temporal adjustments of incoming mossy fiber signals [81,84,90,91,92]. Granule cells in the vestibulocerebellum exhibit spontaneous events resembling bursts of action potentials characteristic of UBC activity in whole-cell recordings, which provide evidence of UBCs as presynaptic elements in the granular layer circuitry (Figure 3A). Additionally, fast EPSPs can be observed in granule cells in response to bursts of action potentials in UBCs following activation of the mossy fiber–UBC synapse (Figure 3B). Furthermore, individual intermediary UBCs have the capacity to modulate the firing phase of their postsynaptic granule cells, with their response to glutamatergic input ranging from partial inhibition (e.g., OFF UBC) to complete excitation (e.g., ON UBC) [82,90,93,94]. The spiking responses in postsynaptic UBCs evoked by mossy fiber stimulation at 100 Hz demonstrated that high-frequency stimulation could produce a burst of spikes that outlasted the stimulus (ON-UBC) or induce a pause in spontaneous action potential firing (OFF-UBC) (Figure 3C).

Thus, UBCs, classified as ON and OFF cells based on their response to glutamatergic input, may offer distinct parallel processing of multisensory input to their targets [90,94,95]. Synaptic pathways involving intermediary UBCs may contribute to long pauses and delays in granule cells, which could be crucial for adaptive learning across a range of time scales, as observed in the cerebellar-like circuits of electric fish [96]. Even in the absence of plasticity, UBCs enable the granular layer to induce delays in the firing rate changes for durations ranging from hundreds to thousands of milliseconds, significantly enhancing the diversity of coding, especially in the temporal domain [92]. While UBCs are distributed across the cerebellum, their prevalence is particularly remarkable in regions governing eye movement and vestibular processing. The UBC network represents a mechanism that may prolong granule cell firing and contribute to the maintenance of the sensory signals underlying motor learning with delayed sensory feedback [84,91,95]. A computational model has explored the implications of the varied phase shifts in UBC responses to sinusoidal vestibular inputs (Figure 4). In the absence of UBCs, granule cells fire in synchrony with extrinsic mossy fiber inputs from the vestibular system, aligning with preferred phases of movement (Figure 4A). However, when UBCs are integrated into the network, the response of granule cells exhibits significantly more varied phase shifts (Figure 4B).

This diversity in the UBC phase shifts results in a range of phase shifts in granule cells. Consequently, this temporal diversity enables the identification of granule cell firing at any phase of a sinusoidal head movement, aligning with adaptive filter models and in vivo observations [97,98,99,100,101]. However, the presence of UBCs in the dorsal vermis and cerebellar hemispheres of higher mammals, including humans [73,102,103,104], underscores their potential role beyond vestibular sensory processing. This suggests new perspectives on UBC involvement in modulating granule cell activity to support complex motor and cognitive tasks as well.

## 5. Granular Layer Plasticity: Insights from Computational Models

The granular layer of the cerebellum exhibits a highly organized arrangement, facilitating the rapid processing of mossy fiber signals within a millisecond timeframe. This intricate architecture gives rise to a diverse spectrum of spatial-temporal patterns through a combination of inhibitory mechanisms and learning processes [56,85,87]. Golgi cells, extending their inhibitory influence across numerous granule cells, establish a brief 5-millisecond window for signal processing (i.e., the time window effect, see [56]), optimizing the transmission of spikes to Purkinje cells [58,105]. Furthermore, bidirectional plasticity at the mossy fiber–granule cell synapse finely regulates the spike timing and number, shaping the window-matching effect due to Golgi cell inhibition [56]. Initial efforts to simulate the impact of granular layer plasticity employed a firing rate model, where the synaptic weights were modulated to optimize the information transfer through the granular layer [106]. This model highlighted the need to balance the strength of mossy fiber synapses with that of Golgi cell synapses, complemented by alterations in the intrinsic excitability, reproducing experimental findings regarding plastic changes in the intrinsic excitability [10]. Additionally, it anticipated the role of acetylcholine in plasticity gating [31] and the role of inhibitory plasticity between Golgi cells and granule cells [107]. Subsequently, recent computational models have simulated the impact of distributed synaptic weights in the cerebellar granular layer network [63,108]. These models elucidated the crucial role of the synaptic weights at various connections in regulating the spike number and positioning in granule cells in response to mossy fiber bursts. Specifically, the synaptic weights at mossy fiber to granule cell synapses regulated the delay of the first spike, while those at mossy fiber and parallel fiber to Golgi cell synapses controlled the duration of the time window for spike emission. Additionally, the weights of synapses governing Golgi cell activation modulated the granule cell inhibition intensity, thereby influencing the spike emission. Through these mathematical models, it was demonstrated that different combinations of synaptic weights optimize either the spike timing precision or spike number, effectively governing transmission and filtering properties along the mossy fiber pathway [63,108]. Furthermore, these models discerned distinct roles for various components of inhibition, with lateral inhibition dictating the center-surround effect, feedforward inhibition influencing the time-windowing effect, and feedback inhibition regulating coherent oscillations [56,108]. Mathematical modeling has significantly influenced research in the field over the past three decades, revealing that various forms of synaptic and intrinsic plasticity at different sites act conjunctively to enable the cerebellar granular layer to function as an adaptive spatial-temporal filter [97,98,109,110]. In this way, synaptic modifications distributed across the network endow the granular layer with the capability to sustain sensorimotor integration and learning, ultimately orchestrating complex motor and cognitive tasks [109,111,112,113,114,115].

## 6. Conclusions

Various forms of synaptic and non-synaptic plasticity have been described in excitatory and inhibitory granular layer neurons. It is therefore likely that learning in the cerebellum emerges as an integrated process involving different synaptic sites that elaborate different components of learning over different time periods. However, how remodeling of the synaptic weights generates the complex properties of cerebellar learning remains to be understood. Mathematical models, which provide increasing detail about the synaptic plasticity at different sites of the network, may help determine the impact of different sites of plasticity on cerebellar learning. These computational approaches have generated several hypotheses, many of which need to be validated by experimental assessment.

## Figures and Tables

**Figure 1 biology-13-00403-f001:**
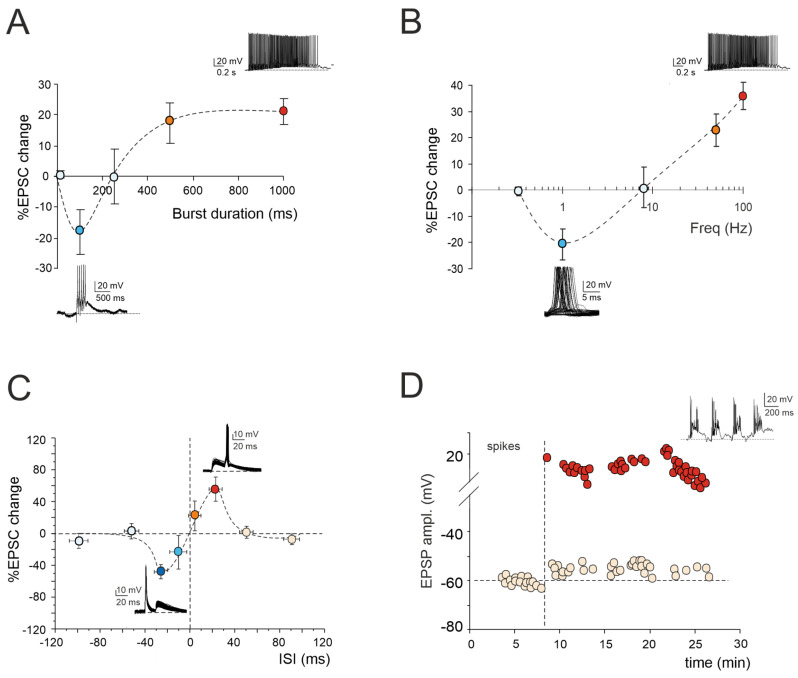
Multiple forms of long-term plasticity at the mossy fiber–granule cell synapse. (**A**) Application of 1–100 continuous mossy fiber stimuli at 100 Hz revealed a net LTD at 100 ms bursts, a neutral point at 250 ms, and an LTP at 500 ms. LTP tends to plateau with trains lasting > 500 ms. The bottom (LTD) and top (LTP) traces show the granule cell response during a 100 ms and 1000 ms burst duration, respectively. Data are reported as mean ± SEM (adapted from [22]). (**B**) LTP or LTD were differentially induced by using the same number of impulses (100) at different frequencies. The probability of obtaining LTD is high at 1 Hz, while that of obtaining LTP is high at 50 Hz, with a neutral point around 8 Hz. The bottom (LTD) and top (LTP) traces show the granule cell response during 100 stimuli at the frequencies of 1 Hz and 100 Hz, respectively. Data are reported as mean ± SEM (adapted from [23]). (**C**) Spike timing-dependent plasticity (STDP) was induced through pairing an excitatory postsynaptic potential (EPSP) evoked by mossy fiber stimulation with a spike elicited by current injection into the granule cell. The spike followed or preceded the onset of the EPSP by Δt = ±5, ±25, ±50, and ±100 ms. Pairing was repeated 60 times at 6 Hz. Average plot represents excitatory postsynaptic current (EPSC) amplitude changes for different Δt. Note the striking transition from maximal LTP to maximal LTD over the narrow time window ~0 ms. The bottom (LTD) and top (LTP) traces show the granule cell response during the STDP induction protocol at Δt = −25 ms and Δt = +25 ms, respectively. Data are reported as mean ± SEM (adapted from [24]). (**D**) Effect of TBS delivered from a holding potential of −50 mV. Exemplar recording showing that LTP is characterized by an EPSP increase leading to spike generation in granule cells. The graph shows the time course of the EPSP amplitude changes and transition to EPSP–spike complexes. The top trace shows the granule cell membrane depolarization elicited by TBS (adapted from [10]).

**Figure 2 biology-13-00403-f002:**
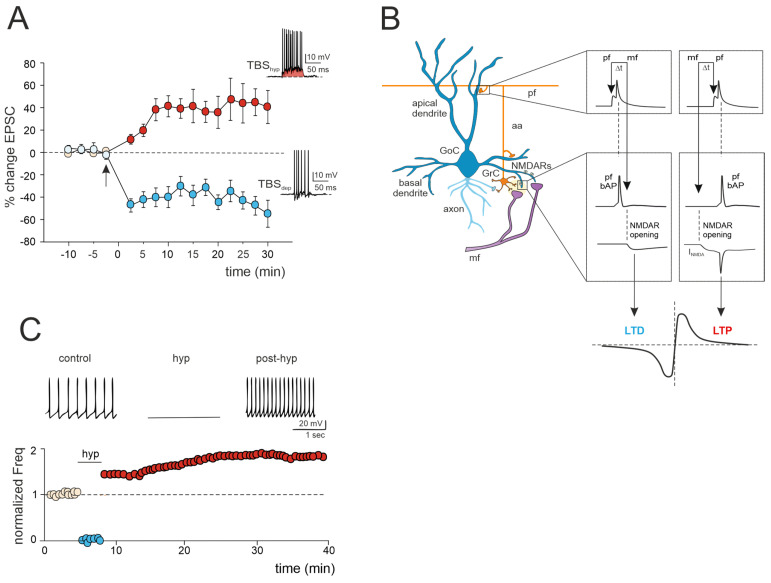
Multiple forms of long-term plasticity at the mossy fiber–Golgi cell synapse. (**A**) Bidirectional plasticity at Golgi cell excitatory synapses: dependence on membrane potential. LTP or LTD is induced by delivering theta burst stimulation (TBS) from different Golgi cell membrane potentials (TBS_hyp_ and TBS_dep_). The graph shows the average time course of the EPSC amplitude changes during LTP and LTD. The arrow indicates the induction time, and each point is the average of 15 contiguous EPSC amplitudes. Data are reported as mean ± SEM. The bottom (LTD) and top (LTP) traces show the Golgi cell response during TBS_hyp_ and TBS_dep_ (adapted from [60]). Note the stronger depolarization and spike generation in TBS_hyp_ (red area) than TBS_dep_. (**B**) STDP at mossy fiber inputs. Backpropagating spikes are elicited by parallel fiber stimulation either ~10 ms before or ~10 ms after a single synapse activation in the mossy fibers. The NMDA receptor-dependent current (INMDA) generated at the mossy fiber synapse is shown in the two cases. The bottom plot shows a theoretical STDP curve (adapted from [61]), with modest INMDA changes leading into the LTD region and large INMDA changes leading into the LTP region. (**C**) Transient hyperpolarization induces a long-term increase in Golgi cell spontaneous firing. Top, example experiment showing that a 3 min negative current injection of −50 pA hyperpolarizes the Golgi cell membrane to −65 mV and suppresses firing activity. An increase in Golgi cell spiking occurred after hyperpolarization. Bottom, Golgi cell shows a long-term increase in spontaneous activity after being hyperpolarized (adapted from [62]).

**Figure 3 biology-13-00403-f003:**
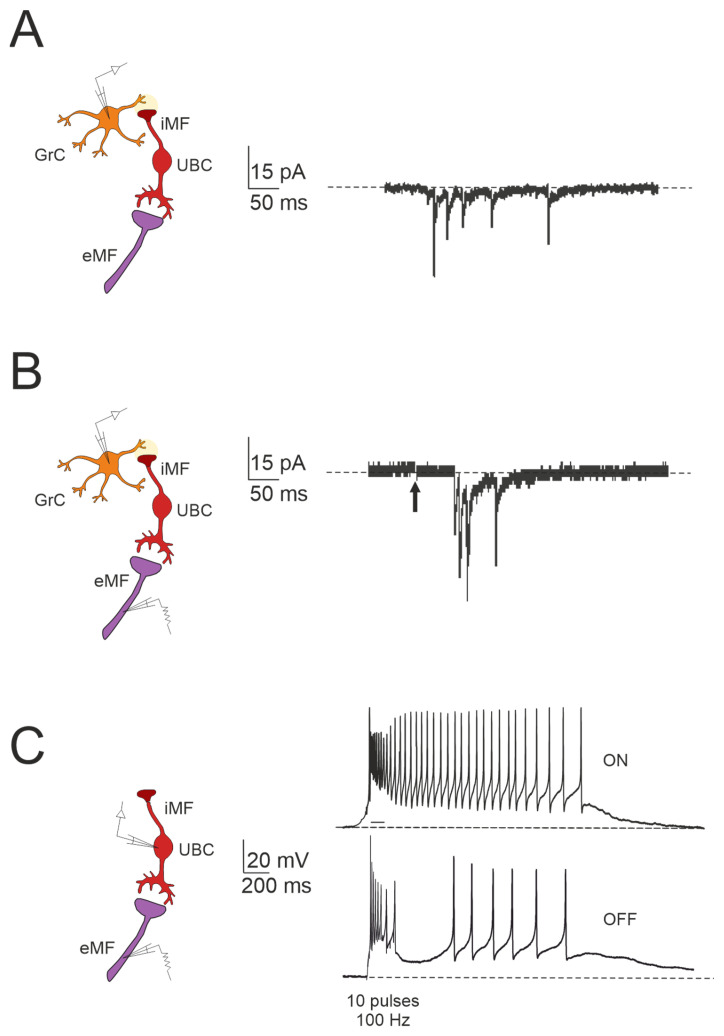
Forward signaling by UBCs in the vestibulocerebellar granular layer. (**A**) Example of whole-cell recordings from a granule cell, displaying spontaneous events reminiscent of UBC discharges (adapted from [84]). (**B**) Burst of EPSCs in a granule cell in response to presynaptic stimulation, with a first peak delay (adapted from [84]). (**C**) Spiking responses in postsynaptic UBC evoked by mossy fiber stimulation (100 Hz): note that high-frequency stimulation can produce a burst of spikes that outlast the stimulus (ON-UBC) or generate a pause in the spontaneous action potential firing (OFF-UBC; adapted from [82]).

**Figure 4 biology-13-00403-f004:**
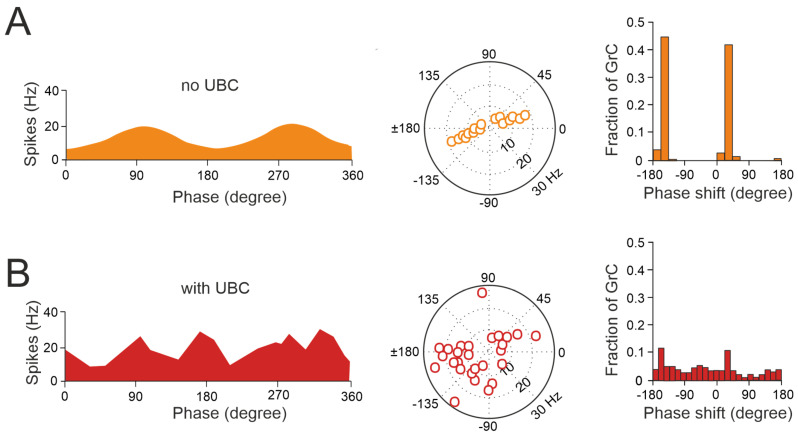
UBCs increase the phase diversity of granule cell responses. (**A**) Simulation of a granular circuit without UBCs. The dynamics of the granular layer model (4500 granule cells receiving inputs from 500 mossy fibers) in response to sinusoidal mossy fiber stimulation, in the presence and absence of UBCs, are simulated. (**A**) Without UBCs, granule cells fire in phase with mossy fiber inputs. Right, average firing rate curve of 30 random selected granule cells in one input period is shown. Left, polar plot and histogram represent the amplitude of modulation and phase shift from the fitted GC firing rate curves, respectively. (**B**) Same as (**A**) but for granular cell outputs with UBCs in the circuit (adapted from [92]).

## Data Availability

No new data were created or analyzed in this study. Data sharing is not applicable to this article.

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
