# Peer review of "Understanding Cerebellar Input Stage through Computational and Plasticity Rules"

_biology, 2024, doi:10.3390/biology13060403_

Round 1
Reviewer 1 Report
Comments and Suggestions for Authors
The most prominent forms of plasticity at excitatory synapses formed by mossy fibers in the cerebellar granular layer are briefly reviewed in this review. The authors provide important insights into how inputs are processed and reconfigured at the cerebellar input stage by highlighting current knowledge of the mechanisms behind synaptic and intrinsic plasticity as well as their functional consequences.
I think this is a very nice paper, despite minor issues that should be addressed:
1. A general English revision is needed.
2. The number of self-citations is excessive. Remove unnecessary ones and reference other authors wherever you can.
3. “Principal neurons” " has a specific meaning (neurons that project outside their layer) and might be misleading here (ABSTRACT)
4. Acronyms should be defined in the text (for example, EPSP, EPSC, TBS, GCL etc…)
5. “This indicates that receptor pathways activated by diverse afferent patterns via calcium concentration changes shared plastic mechanisms to promote the reconfiguration of inputs within the granular layer” (1.1 PARAGRAPH). This is not clear. Are the afferent patterns meaning the different induction patterns?
6. “The balance between LTP and LTD at the mossy fiber-granule cell synapse is regulated not only by specific input patterns but also by molecular factors and neuromodulators. NO is released in the granular layer upon high-frequency mossy fiber stimulation through NMDAR-dependent and NOS-dependent mechanisms. Functioning as a retrograde messenger, NO enhances presynaptic release probability, thereby favoring LTP over LTD. Additionally, activation of α7 nicotinic acetylcholine receptors (α7nAchRs) on both mossy fiber terminals and granule cell dendrites amplifies Ca2+ influx at postsynaptic sites” (1.1 PARAGRAPH). All this block is repeated (almost) identical immediately above.
7. “A recent finding has revealed that the interplay between synaptic response (excitatory postsynaptic potential, EPSP) and spikes can drive STDP at the mossy fiber-granule cell synapse” (1.2 PARAGRAPH). Something to clarify that these are spikes induced in the granule cells without synaptic activation.
8. Phasic and tonic inhibition (2 PARAGRAPH): phasic and tonic inhibition are the two general ways in which inhibition controls granule cell activity. I would not use this distinction as related to feedforward inhibition.
9. It appears insufficient to explain the feedback loops to those who are not familiar of the cerebellar circuit (2 PARAGRAPH).
10. “The characteristics of postsynaptic receptors, combined with limitations on the diffusion of glutamate caused by synaptic ultrastructure and glutamate transporters, thus shape the time course of the resulting steady-state current” (3 PARAGRAPH). Please rephrase, a little difficult to read.
11. “presynaptic inputs” (3.1 PARAGRAPH). Presynaptic respect to the granule cell or the UBC?
Comments on the Quality of English LanguageA moderate revision is needed
Reviewer 2 Report
Comments and Suggestions for Authors
The authors described in detail the implications of synaptic plasticity of the cerebellar microcircuit to realize a realistic computational model. The topic is intriguing and well characterized.
Some issues have been detected, please ameliorate the following points to improve the understanding of the text:
1) The text between lines 114 to 121 is repeated in subsequent lines 122 to 129. I assume this could be an inattentive error. It would be better to delete lines 122 – 129 and save the previous one, since the bibliography is inserted correctly in that part.
2) In the statement from line 130 to the end of section 1.1 the appropriate citation is missing, cit. [31] seems appropriate for the content, but is not placed in the correct place. Furthermore, the following sentence is not clear à “… transmission is abolished inhibiting NO of NOS abolishes, …”. So, to improve the understanding of the text it would be better to add the appropriate reference and improve the sentence.
3) In the paragraph “2. Plasticity in Golgi cells” the abbreviations (GCL, GrC and GoCS) are not explained earlier in the text. Please write the full name before the acronym: Granule cell layer (GCL), cerebellar granule cell (GrC) and cerebellar Golgi cells (GoCs).
4) In section 3.1, lines 353, 355 and 361 refer to figures 3A, B and C respectively. However, it may be appropriate to also cite the relevant references in the text: [81] for lines 353 and 355, [82] for line 361.
5) In the last section “4. Plasticity of the granular layer: insights from computational models” in line 437 the following statement “… to address the absence of teaching lines…” does not appear very clear, do you mean “… learning lines…”? Please clarify this sentence.
Comments on the Quality of English LanguageThere are minimal typing errors in the text, for example in line 212 “…broad pf convergence…” à “…broad of convergence…”. Please check for other possible mistakes.
Round 2
Reviewer 1 Report
Comments and Suggestions for Authors
The authors addressed all of the issues raised and the paper can now be accepted for publication
Reviewer 2 Report
Comments and Suggestions for Authors
Dear authors,
Thank you for your response and comments on the proposed problems. The text appears clearly improved. There were some minor typos as line 58 is missing a dot in the sentence "... pathway [12,13] Secondly, activation...", please check for others. Finally, the acronym EPSD is defined in line 323, but not in line 139 where it first appears in the text.